# FTF-ER: Feature-Topology Fusion-Based Experience Replay Method for Continual Graph Learning

## ABSTRACT

Continual graph learning (CGL) is an important and challenging task that aims to extend static GNNs to dynamic task flow scenarios. As one of the mainstream CGL methods, the experience replay (ER) method receives widespread attention due to its superior performance. However, existing ER methods focus on identifying samples by feature significance or topological relevance, which limits their utilization of comprehensive graph data. In addition, the topology-based ER methods only consider local topological information and add neighboring nodes to the buffer, which ignores the global topological information and increases memory overhead. To bridge these gaps, we propose a novel method called Feature-Topology Fusion-based Experience Replay (FTF-ER) to effectively mitigate the catastrophic forgetting issue with enhanced efficiency. Specifically, from an overall perspective to maximize the utilization of the entire graph data, we propose a highly complementary approach including both feature and global topological information, which can significantly improve the effectiveness of the sampled nodes. Moreover, to further utilize global topological information, we propose Hodge Potential Score (HPS) as a novel module to calculate the topological importance of nodes. HPS derives a global node ranking via Hodge decomposition on graphs, providing more accurate global topological information compared to neighbor sampling. By excluding neighbor sampling, HPS significantly reduces buffer storage costs for acquiring topological information and simultaneously decreases training time. Compared with state-of-the-art methods, FTF-ER achieves a significant improvement of 3.6% in AA and 7.1% in AF on the OGB-Arxiv dataset, demonstrating its superior performance in the class-incremental learning setting.

## CCS CONCEPTS

• **Mathematics of computing → Graph algorithms**; • **Computing methodologies → Neural networks**.

## KEYWORDS

Continual Graph Learning, Experience Replay, Hodge Decomposition

## 1 INTRODUCTION

Continual learning (CL) in the field of graph neural networks (GNNs) [12, 17, 30] has become increasingly significant due to

*ACM MM, 2024, Melbourne, Australia*
© 2024 Copyright held by the owner/author(s). Publication rights licensed to ACM.
ACM ISBN 978-x-xxxx-xxxx-x/YY/MM
https://doi.org/10.1145/nnnnnnn.nnnnnnn

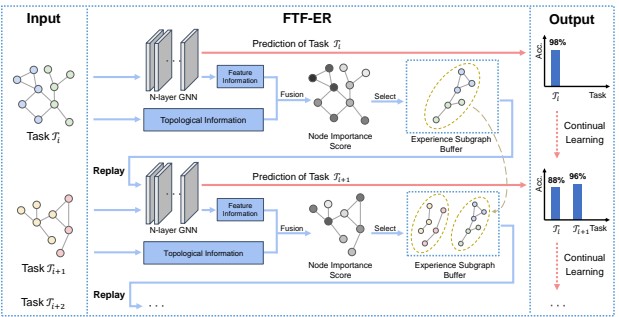

**Figure 1: FTF-ER workflow: Nodes are selected from each class based on importance scores to create induced subgraphs for the buffer. This enables the model to maintain classification performance across tasks, i.e., continual learning.**

its potential applications in dynamic and evolving environments. Traditional GNN models are designed to operate on static graph data, making them inadequate for scenarios where graphs evolve over time. Continual graph learning (CGL) addresses this challenge by extending GNNs to adapt to changing graph structures and data distributions [36]. This enables models to continuously learn and update their knowledge without forgetting previous information. The ability to learn from sequential data and adapt to new dynamic information is crucial for various real-world applications such as social network analysis [13], recommendation systems [11] and traffic prediction [7]. In this context, developing effective continual learning strategies for graphs has emerged as a key research area with the aim of enhancing the robustness and adaptability of GNN models in dynamic environments.

CGL methods can be categorized into three classes [47]: regularization methods, parametric isolation methods and experience replay (ER) methods. Among them, ER methods, inspired by the complementary learning systems theory in cognitive science [19, 25], have achieved SOTA performance in both traditional CL and CGL fields. To help the model stay good at the earlier tasks, ER methods use a partial collection of training samples from previous tasks and reintroduce them when training on new tasks. In recent years, some studies integrate ER methods to deal with the issue of graph learning in streaming scenarios [36]. ER-GNN [51] stores representative nodes at the feature level for experience replay. SGNN-GR [37] incorporates generative replay to learn and generate fake historical samples. Recently, SSM [49] stores sparsified subgraphs to gain topological-level experience and achieves SOTA performance in CGL. However, the ER methods mentioned above primarily select impactful samples based on either feature significance or topological relevance, which limits their utilization of the comprehensive graph data. Furthermore, to provide local topological information, SSM expands the buffer by adding nodes that are not present in the experience samples, resulting in the neglect of global topological information and the requirements of additional storage.

To address the aforementioned issues, we integrate information from both the feature and topological levels and propose a novel experience replay method called Feature-Topology Fusion-based Experience Replay (**FTF-ER**) to mitigate the catastrophic forgetting issue in CGL. Figure 1 shows the workflow of FTF-ER, and the essence of the workflow centers on the approach to acquiring information for sampling. Specifically, we employ two theoretically grounded approaches to calculate the feature-level and topological-level importance scores of nodes. Firstly, inspired by [26], we use gradient norm score (GraNd) to compute the node importance in the feature space. GraNd measures the importance of a node by evaluating the change in the loss of all other nodes when the node is added to the training set. Furthermore, we propose Hodge potential score (HPS) as a novel module to calculate the topological importance of nodes by using Hodge decomposition on graphs (HDG) [15, 21]. HDG is an effective tool for studying the topological properties of nodes that can establish a natural and computable node potential function. This function can derive a global ranking from pairwise comparison relationships among all nodes. It explicitly highlight the global topological importance of each node. To the best of our knowledge, our study represents the first application of HDG to node importance computation. By precomputing the HPS of nodes before training, our method efficiently captures comprehensive global topological information in the node sampling phase. HPS enhances the accuracy of topological information while eliminating the need for additional nodes during the subgraph creation phase.

In summary, our contributions can be outlined as follows:

- We present Feature-Topology Fusion-based Experience Replay (**FTF-ER**), a novel framework from an overarching standpoint, which aims to efficiently and effectively alleviate the catastrophic forgetting problem in continual graph learning.
- To fully utilize the comprehensive graph data, we propose an integrative approach to better represent node importance. We first normalize the node importance scores on features and topology separately, and then obtain the aggregated importance by calculating the weighted average of two scores, thereby improving the accuracy of node importance.
- We propose Hodge potential score (HPS), a preprocessing module to capture global topological information without adding neighboring nodes to the buffer. Consequently, HPS further utilizes global topological information and reduces buffer storage costs of topology-based ER methods.
- We conduct extensive experiments on four mainstream graph datasets, and achieve state-of-the-art performance on accuracy, time efficiency in the class-incremental learning setting. Moreover, despite leveraging topological information, our buffer storage costs are comparable to topology-agnostic methods.

## 2 RELATED WORK

### 2.1 Experience Replay

Continual graph learning seeks to sequentially train the model as graph data from various tasks are received in a stream. Similar to continual learning research in other fields [14, 28, 40], research methods for CGL can be mainly categorized into three major types [9, 46]. The first family consists of regularization methods that focus on preserving the parameters inferred in one task while training on

another [1, 8, 18, 20, 22]. The second family is parametric isolation methods that separate parameters from different tasks explicitly [29, 39, 43, 44, 48]. The last family is experience replay methods that store a limited amount of representative data from previous tasks and replay them during training on subsequent tasks in order to maintain the model's ability to classify past tasks [2, 23, 27]. As an illustration, Gradient Episodic Memory (GEM) [23] constrains the gradient of the current task to prevent an increase in the loss associated with the data stored in episodic memory. In the domain of graphs, Experience Replay Graph Neural Network (ER-GNN) [51] stores representative nodes at the feature level for experience replay, while Sparsified Subgraph Memory (SSM) [49] stores sparsified subgraphs that include $k$-hop neighboring nodes to gain local topological-level experience. However, previous methods primarily have two drawbacks: (1) They select samples based on either feature significance or topological relevance, limiting their comprehensive utilization of graph data. (2) SSM adds neighboring nodes to the buffer, overlooking global topological information and increasing storage needs. Different from them, our FTF-ER combines both feature and topological information of nodes to provide an overall perspective for alleviating the catastrophic forgetting problem. Furthermore, we propose Hodge potential score to obtain global topological information without introducing extra nodes, thereby further utilizing global topological information and reducing the buffer storage overhead of topology-based ER methods.

### 2.2 Application of Hodge Decomposition

In the realm of machine learning, Hodge decomposition has found several innovative applications, serving as a foundational tool for understanding complex data structures through the lens of algebraic topology [5, 31, 32]. For example, previous work [34] develops a set of tools for analyzing 3D discrete vector fields on tetrahedral grids using a Hodge decomposition approach. In addition, researchers apply Hodge decomposition to robustly find global rankings in the presence of outliers for image processing [45]. In the field of robotics, Helmholtz-Hodge decomposition is used to create algorithms that approximate incompressible flows for agent control and stream function computation in graph-modeled environments [16]. In [15] and [21], researchers introduce Hodge decomposition on graphs (HDG) and apply it to statistical ranking problems. Furthermore, in [42], researchers use HodgeRank for paired comparison data in the multimedia domain, assessing video quality and analyzing inconsistencies. For the advantage of Hodge decomposition's ability to reliably and globally rank nodes in graphs, even among irregular data, it has seen widespread application in various fields. To the best of our knowledge, we first introduce HDG to CGL, thus obtaining global topological information and reducing buffer storage requirements of topology-based ER methods.

## 3 METHOD

### 3.1 Preliminaries

**Notations.** For a graph $\mathcal{G} = (\mathcal{V}, \mathcal{E})$ with $n$ nodes, we have a node set $\mathcal{V} = \{v_1, v_2, \ldots, v_n\}$ and an edge set $\mathcal{E} = \{(u, v) | u \in \mathcal{V}, v \in \mathcal{V}\}$. We use the adjacency matrix $\mathbf{A} \in \mathbb{R}^{n \times n}$ to describe the connectivity of $\mathcal{G}$, and each non-zero element in $\mathbf{A}$ corresponds to an edge in $\mathcal{E}$. In this paper, we study the node classification problem in

**Figure 2: The complete node importance score calculation process for our FTF-ER. In the schematic diagram of Potential, the grayscale of nodes represents their importance, and the edges are only used to indicate the comparative results of importance. In the Sample Score stage, before mixing the two scores, they need to be normalized separately as shown in Eq. (18).**

which each node $v \in \mathcal{V}$ has a category label $y^l \in \mathcal{Y}$ where $\mathcal{Y} = \{y^1, y^2, \ldots, y^c\}$ is the label set and $c$ is the number of classes. GNNs are the mainstream solution for node classification problems.

**Problem Definition.** In this paper, we focus on the continual graph node classification problem. In a learning process, the model is continually trained on a sequence of disjoint tasks $\mathcal{T} = \{\mathcal{T}_1, \mathcal{T}_2, \ldots, \mathcal{T}_K\}$, where $K$ represents the number of tasks. Each task $\mathcal{T}_i$ comprises multiple non-overlapping classes $\mathcal{Y}_i = \{y^1, y^2, \ldots, y^{c_i}\}$ and $c_i$ is the number of classes in task $\mathcal{T}_i$. We define the node set of $\mathcal{T}_i$ as $\mathcal{V}_i = \{v | y(v) \in \mathcal{Y}_i, v \in \mathcal{V}\}$, where $y(v)$ is the label of node $v$. Then the induced subgraph of $\mathcal{T}_i$ can be represented as $\mathcal{G}_i = (\mathcal{V}_i, \mathcal{E}_i), \mathcal{E}_i = \{(u, v) | u, v \in \mathcal{V}_k, (u, v) \in \mathcal{E}\}$. In the continual learning settings, different tasks correspond to different induced subgraphs without overlap on $\mathcal{Y}$. Once the learning of a task is completed, the vast majority of the training data related to this task are no longer available. The goal of CGL is to achieve consistent high performance across all tasks in the sequence, addressing both current task performance and mitigating the catastrophic forgetting problems for past tasks.

**Incremental Learning Settings.** Continual learning has two main settings: task-incremental learning (task-IL) and class-incremental learning (class-IL). The key difference lies in whether task indicators are provided to the model during testing. In task-IL, the model receives a task indicator, enabling it to concentrate solely on the classes relevant to the current task during classification. Conversely, in class-IL, the model must identify all previously learned classes without the assistance of task indicators. Class-IL is considered more challenging due to the larger classification dimension and the absence of explicit task boundaries. Previous studies [47] have shown that CGL methods significantly perform better in task-IL than in class-IL. In this paper, we aim to tackle the more challenging class-IL tasks to showcase the effectiveness of our method.

## 3.2 Experience Replay on Subgraphs

In the field of graphs, due to the presence of rich topological structures, experience replay (ER) methods can be extended to ER on subgraphs. The traditional ER methods preserve a small number of training samples from past tasks and replay them during the subsequent tasks training in order to retain the model's classification ability for past tasks. The key to ER methods lies in designing a rational experience sample selection strategy. Previous research has revealed that samples are inherently unequal; some samples may carry information that can significantly improve model performance, while the addition of other samples may have little impact on enhancing model effectiveness [33]. For a given task $\mathcal{T}_k$ and its training dataset $\mathcal{D}_k^{tr}$, ER methods use an experience sample strategy $\mathcal{S}(\cdot)$ to collect experience samples from $\mathcal{D}_k^{tr}$ and add them to the experience buffer $\mathcal{B}$. A simple way to replay the experience in $\mathcal{B}$ is applying an auxiliary loss:

$$\mathcal{L} = \sum_{x_i \in \mathcal{D}_k^{tr}} l(f(x_i; \boldsymbol{\theta}), y_i) + \lambda \sum_{x_j \in \mathcal{B}} l(f(x_j; \boldsymbol{\theta}), y_j), \quad (1)$$

where $l(\cdot, \cdot)$ denotes the loss function, $f(\cdot; \boldsymbol{\theta})$ denotes the model training on the task sequence, and $\lambda$ is utilized to balance the auxiliary loss. When training $f(\cdot; \boldsymbol{\theta})$ in $\mathcal{T}_k$, $\mathcal{B}$ contains the experience samples collected from $\mathcal{T}_1$ to $\mathcal{T}_{k-1}$.

For GNNs, which have the capability to effectively harness and leverage the rich topological information embedded within the graph structure, the selection process of experience samples should consider not only the importance of samples in isolation but also their significance and influence at the topological level. A naive approach to incorporating topological information into experience is to sample nodes and their induced subgraph simultaneously:

$$(\mathcal{V}_k^{buf}, \mathcal{G}_k^{buf}) = \mathcal{S}(\mathcal{D}_k^{tr}, \mathcal{G}_k), \quad (2)$$

where $\mathcal{G}_k^{buf} = \mathcal{G}_k(\mathcal{V}_k^{buf})$ is the induced subgraph of $\mathcal{G}_k$ with respect to the experience node set $\mathcal{V}_k^{buf}$. And $\mathcal{S}(\cdot)$ denotes the strategy used to select nodes and generate subgraph. With the assistance of the topological information carried by the induced subgraph, the loss function for CGL can be defined as follows:

$$\mathcal{L} = \sum_{x_i \in \mathcal{D}_k^{tr}} l(f(x_i, \mathcal{G}_k; \boldsymbol{\theta}), y_i) + \lambda \sum_{x_j \in V(\mathcal{B})} l(f(x_j, \mathcal{G}_\mathcal{B}; \boldsymbol{\theta}), y_j), \quad (3)$$

where $\mathcal{G}_\mathcal{B}$ is the subgraph stored in the buffer $\mathcal{B}$ and $V(\mathcal{B})$ denotes the node set stored in $\mathcal{B}$. In other topology-based ER methods, $\mathcal{G}_\mathcal{B}$ is not necessarily an induced subgraph of $V(\mathcal{B})$, without loss of generality. For example in SSM [49], to obtain more topological information, the node set of $\mathcal{G}_\mathcal{B}$ includes not only $V(\mathcal{B})$ but also a subset of $\mathcal{N}(v)$ for each node $v$ in $V(\mathcal{B})$, where $\mathcal{N}(v)$ represents the set of neighbors of node $v$ in the graph.

In this paper, due to the acquisition of additional topological information reflected in the selection strategy of $V(\mathcal{B})$, we simply let $\mathcal{G}_\mathcal{B} = \mathcal{G}(V(\mathcal{B}))$. Therefore, we do not need to bear the storage requirements of non-sample nodes in $\mathcal{G}_\mathcal{B}$. The pipeline of the method we propose is illustrated in Figure 1.

### 3.3 Node Importance Score

To introduce our experience node selection strategy that integrates feature information and topological information, we first adopt the gradient norm (**GraNd**) score [26], which is used to compute the importance of nodes in the feature space. Next, we propose Hodge potential score (**HPS**) to calculate the importance scores of nodes at the topological level by HDG. Finally, we integrate the two importance scores to a weighted average score and provide the mixed node importance score for node selection. The complete computational process for calculating the node importance scores we propose is depicted in Figure 2.

*3.3.1 **Gradient Norm Score for Feature Importance**. Inspired by [26], we introduce gradient norm score (**GraNd**) to measure the importance of nodes at the feature level. Intuitively, we can define a sample's importance as its contribution to minimizing the model's loss function during training. Put simply, a sample is considered important if it helps reduce the loss of other samples when the model parameters are optimized using this particular sample. The importance of the training sample $x_i$ can be formalized as follows:

$$\mathcal{I}_i = \sum_{x_j \in X^{tr} - \{x_i\}} (l(f(x_j; \boldsymbol{\theta}), y_j) - l(f(x_j; \boldsymbol{\theta}'), y_j)), \quad (4)$$

where

$$\boldsymbol{\theta}' = \boldsymbol{\theta} - \eta \nabla_{\boldsymbol{\theta}} l(f(x_i; \boldsymbol{\theta}), y_i), \quad (5)$$

and $\eta$ denotes the learning rate.

This definition intuitively represents the generalization ability of the selected sample, essentially measuring the sample's value in improving model predictions on other data. Simply put, we can estimate a sample's importance by calculating the gradient's norm it produces. In DNN models, updated via gradient descent, this gradient norm mirrors its effect on adjusting model parameters. Through theoretical validation [26], researchers approximate this effect by calculating the $L_2$ norm of the model gradients after training the respective node. GraNd is formally defined as follows:

$$\mathbf{S}^{loss} = [S_1^{loss}, S_2^{loss}, \dots, S_{|X_k^{tr}|}^{loss}],$$

$$S_i^{loss} = \mathbb{E}_{\boldsymbol{\theta}} ||\nabla_{\boldsymbol{\theta}} l(f(x_i; \boldsymbol{\theta}), y_i)||_2. \quad (6)$$

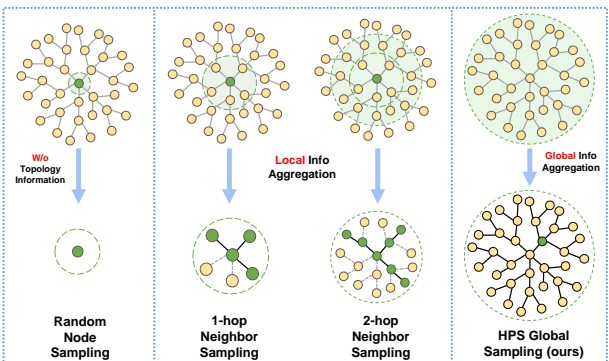

**Figure 3: Acquiring topological information by random node sampling, SSM [49] with 1-hop, 2-hop neighbor sampling, and FTF-ER (ours). Green nodes indicate the ones selected to be added to the buffer, and bold solid lines represent the information flow between nodes during the sampling process.**

The contribution of GraNd lies in its ability to make reasonable judgments on the importance of samples early in the training process, which aligns with the setting of incremental learning in CL. Therefore, we choose GraNd to calculate the importance of samples at the feature level, with the expectation that the selected samples can best fit the distribution of the entire dataset.

*3.3.2 **Hodge Potential Score for Topology Importance**. We introduce the Hodge Potential Score (**HPS**), a novel measure of a node's topological importance in a graph. We discuss the motivation behind proposing HPS. Unlike other topology-based ER methods or the commonly used PageRank [3] algorithm that acquire local topological information through neighboring node sampling, our HPS module uses the global ranking derived by Hodge decomposition on graphs (HDG) to measure the global topological importance of nodes, which can achieve more accurate global topological information. Additionally, in CGL, the storage overhead during model execution is also a crucial metric for ER methods. Applying HPS can circumvent the drawback of previous ER methods requiring the storage of neighboring nodes to obtain topological information.

To utilize HDG, we first introduce several key definitions of HDG used in our method. The complete descriptions of HDG is presented in *Appendix*. Let $\Omega^k(M)$ be a $k$-form on an $n$-dimensional smooth manifold $M$, $d$ be the exterior derivative operator, and $\delta$ be the adjoint map of $d$, we provide the following definitions:

*Definition 3.1 (Hodge Potential Score).*
$$\Omega^0(\mathcal{G}) \triangleq \{s : \mathcal{V} \mapsto \mathbb{R}\}. \quad (7)$$

*Definition 3.2 (Edge Flows).*
$$\Omega^1(\mathcal{G}) \triangleq \{X : \mathcal{V} \times \mathcal{V} \mapsto \mathbb{R} | X(i, j) = -X(j, i), (i, j) \in \mathcal{E}\}. \quad (8)$$

*Definition 3.3 (Gradient Operator).* Let **grad** be the gradient operator, $s_i, s_j \in \Omega^0(\mathcal{G})$, we have
$$(d_0 s)(i, j) \triangleq (\mathbf{grad}\, s)(i, j) \triangleq s_j - s_i, (i, j) \in \mathcal{E}. \quad (9)$$

*Definition 3.4 (Negative Divergence Operator).* Let $w$ be the weight of an element in $\Omega^k(\mathcal{G})$, $w_i \in \Omega^0(\mathcal{G})$, $w_{ij} \in \Omega^1(\mathcal{G})$ and **div** be the divergence operator, we have
$$(\delta_0 X)(i) \triangleq (-\mathbf{div}\, X)(i) = -\sum_j \frac{w_{ij}}{w_i} X(i, j). \quad (10)$$

*Definition 3.5 (Graph Laplacian Operator).*

$$\Delta_0 \triangleq \delta_0 d_0 \triangleq -\mathbf{div}(\mathbf{grad}). \tag{11}$$

To calculate HPS, we primarily make use of Definition 3.5. Definition 3.1 defines a function on graphs that maps a set of nodes $\mathcal{V}$ to the real number field $\mathbb{R}$. This function is naturally suitable as a node importance scoring function and is referred to as Hodge potential score. Inspired by [15], we calculate HPS as follows:

$$\Delta_0 s = -\mathbf{div}\, \overline{Y}, \tag{12}$$

where $\overline{Y}$ denotes inconsistent local rankings, which is consistent with the **grad** in Definition 3.5. In CGL, we simplify it to an anti-symmetric adjacency matrix $\overline{\mathbf{A}}$ on directed graphs, where

$$\overline{\mathbf{A}}_{ij} = \begin{cases} 1, & \text{if } (i, j) \in \mathcal{E} \text{ and } (j, i) \notin \mathcal{E}, \\ -1, & \text{if } (j, i) \in \mathcal{E} \text{ and } (i, j) \notin \mathcal{E}, \\ 0, & \text{otherwise}. \end{cases} \tag{13}$$

For undirected graphs, $\overline{\mathbf{A}} = \mathbf{A}$.

The minimum norm solution of Eq. (12) is

$$s* = -\Delta_0^\dagger \mathbf{div}\, \overline{\mathbf{A}}, \tag{14}$$

where $\dagger$ indicates a Moore-Penrose inverse.

By applying Definition 3.4, we have

$$\mathbf{div}\, \overline{\mathbf{A}} = \delta \overline{\mathbf{A}} = \overline{\mathbf{A}} \cdot [1, \cdots, 1]^T. \tag{15}$$

And we have the common definition of the graph Laplacian:

$$\Delta_0 = \mathbf{D} - \mathbf{A}, \tag{16}$$

where $\mathbf{D} = \text{diag}(\deg(1), \cdots, \deg(n))$ and $\deg(i)$ denotes the degree of node $v_i$.

By combining the above equations, we consolidate the formula for calculating HPS as follows:

$$\mathbf{S}^{topo} = s* = -\Delta_0^\dagger \delta \overline{\mathbf{A}}. \tag{17}$$

It's obvious that for any node, HPS aggregates information from all other nodes. Figure 3 shows that our proposed HPS module achieves global topological information aggregation during node sampling, enhancing the accuracy of topological information and reducing buffer storage costs compared to the existing topology-based ER methods. Additionally, it is worth noting that the process of calculating HPS can be considered as a data preprocessing step, where the HPS of the various subdatasets after task partitioning can be computed before training. (Alternatively, HPS can be computed only once for the complete dataset.) This enables our approach to eliminate the need for training time when obtaining topological information, boosting training efficiency.

### 3.3.3 *Fusion of GraNd and HPS*.

To fully leverage the comprehensive graph data, we integrate GraNd and HPS as follows. We adopt a weighted average approach to combine node importance scores $\mathbf{S}^{loss}$ and $\mathbf{S}^{topo}$. Due to the different scales of the two types of scores $\mathbf{S}$, it is necessary to perform min-max normalization on each of them before the combination:

$$\text{norm}(\mathbf{S}) = \frac{\mathbf{S} - \min(\mathbf{S})}{\max(\mathbf{S}) - \min(\mathbf{S})}. \tag{18}$$

Then we define the mixed node importance score as follows:

$$\mathbf{S}^{mix} = (1 - \beta)\, \text{norm}(\mathbf{S}^{loss}) + \beta\, \text{norm}(\mathbf{S}^{topo}), \tag{19}$$

where $\beta \in [0, 1]$ is a hyper-parameter used to adjust the emphasis of sampling. A higher $\beta$ value indicates a stronger emphasis on the topological importance of nodes during sampling. Due to variations in the characteristics of different datasets, different $\beta$ values are

---

**Algorithm 1** Framework of our FTF-ER.

---

**Input:** Task sequence $\mathcal{T} = \{\mathcal{T}_1, \mathcal{T}_2, \ldots, \mathcal{T}_K\}$; Experience buffer $\mathcal{B}$; Number of sampled nodes for each class $b$.

**Output:** A model $f(\cdot; \theta)$ that performs well on all tasks.

1: $\mathcal{B} \leftarrow (\emptyset, \emptyset)$
2: Initialize $\theta$ at random
3: **for** $\mathcal{T}_i$ in $\mathcal{T}$ **do**
4:      Obtain training dataset $\mathcal{D}_i^{tr} = (\mathcal{V}_i^{tr}, \mathcal{G}_i)$ from $\mathcal{T}_i$
5:      Extract experience nodes $V(\mathcal{B})$ and subgraph $\mathcal{G}_\mathcal{B}$ from $\mathcal{B}$
6:      Compute $\mathcal{L}(f(\cdot; \theta), \mathcal{D}_i^{tr}, V(\mathcal{B}), \mathcal{G}_\mathcal{B})$ using Eq. (3)
7:      $\theta \leftarrow arg\, min_\theta(\mathcal{L})$
8:      Compute $\mathbf{S}^{mix}(f(\cdot; \theta), \mathcal{D}_i^{tr})$ using Eq. (19)
9:      $\mathcal{V}_i^{buf} \leftarrow Select(\mathcal{V}_i^{tr}, \mathbf{S}^{mix}, b)$
10:      $\mathcal{B} \leftarrow (V(\mathcal{B}) \cup \mathcal{V}_i^{buf}, \mathcal{G}_\mathcal{B} \cup \mathcal{G}_i(\mathcal{V}_i^{buf}))$
11: **end for**

---

used during experimentation. A more detailed analysis of the values for $\beta$ is presented in Section 4.5.

After calculating the mixed node importance scores, we can use these scores through two strategies: deterministic sampling or probabilistic sampling. The deterministic strategy directly sorts the nodes based on their scores and selects the top $b$ nodes as the experience node set. In contrast, the probabilistic strategy uses the scores after standardization as a probability mass function: $p(i) = \frac{S_i^{mix}}{\sum_{j=0}^{|X_k^{tr}|} S_j^{mix}}$. The experience node set is obtained by performing $b$ rounds of sampling without replacement from the probability distribution defined by $p(i)$.

In conclusion, the experience selection strategy we propose can be described as follows: for each class in $\mathcal{T}_k$, we sample $b$ nodes along with their induced subgraph $\mathcal{G}_k(\mathcal{V}_k^{buf})$, and then add them to buffer $\mathcal{B}$. To obtain the most accurate importance scores, we calculate the $\mathbf{S}^{loss}$ of $\mathcal{T}_k$ and perform node selection after the completion of training for $\mathcal{T}_k$, as demonstrated in Algorithm 1.

## 3.4 Algorithm Complexity Analysis

**Time Complexity Analysis.** To demonstrate the validity of our experiments regarding the algorithm's runtime costs, we analyze the time complexity of FTF-ER from a complexity theory perspective. The majority of the time cost in our FTF-ER is concentrated in the calculation of HPS and GraNd. The calculation of HPS is completed in the preprocessing stage and does not contribute to the runtime cost. Besides, GraNd's computation process requires a calculation for each node to be sampled, resulting in a complexity of $O(n)$ when the number of sampled nodes is $n$, similar to ER-GNN. During the sampling stage, both FTF-ER and ER-GNN perform single sampling based on importance scores, resulting in a complexity of $O(n)$. For the random neighbor sampling version of SSM [49], there is no importance score calculation stage. However, due to the need to sample neighbors for each node, the time complexity of the sampling stage in SSM is $O(n^2)$. In summary, the total time complexity of FTF-ER and ER-GNN is $O(n) + O(n) = O(n)$, while the total time complexity of SSM is $O(n^2)$. This explains why the

**Table 1: Statistical information of four public graph datasets.**

| Dataset | Amazon Computers [24] | Corafull [4] | OGB-Arxiv [38] | Reddit [12] |
|---|---|---|---|---|
| # nodes | 13,381 | 18,800 | 169,343 | 232,965 |
| # edges | 491,556 | 125,370 | 2,315,598 | 114,615,892 |
| # classes | 10 | 70 | 40 | 40 |
| # tasks | 5 | 35 | 20 | 20 |

time cost of SSM is significantly higher than that of FTF-ER and ER-GNN during the experimental process.

**Space Complexity Analysis.** To validate the correctness of our experiments concerning buffer storage overhead, we analyze the space complexity of the buffer in FTF-ER from a complexity theory perspective. Similar to SSM [49], when the number of sampled nodes is $n$, the space complexity of the buffer in FTF-ER is $O(n)$. However, since FTF-ER does not require introducing additional neighboring nodes, the additional space complexity of the buffer is $O(1)$, while the additional space complexity of SSM remains $O(n)$. Thus, we theoretically demonstrate that the space occupancy of our FTF-ER buffer is lower than that of SSM.

## 4 EXPERIMENTS

### 4.1 Experimental Details

**Datasets.** We investigate multiple public datasets in the CGL field and select the four most representative graph node classification datasets for experimental exploration. These datasets have a wide coverage in terms of scale, content, and structure, enabling an effective evaluation of the generalizability of various CGL methods. To enhance the difficulty of the experiments, following the setting of [47], we set the number of classes for each task to 2 on all datasets, thereby maximizing the length of the task sequence. In addition, to adapt to the majority of GNN backbones designed for undirected graphs, we standardize all datasets (including undirecting, removing weights, eliminating self-loops, and extracting the largest connected component). For each class, we divide the data into training, validation, and test sets in a ratio of 6:2:2. The statistical information of all the datasets is shown in Table 1.

**Evaluation Metrics.** There are two main types of evaluation metrics for CGL: average performance (AP) and average forgetting (AF) [23]. AP is used to measure the average testing performance of the model across all tasks, while AF can quantify the degree of forgetting on previously learned tasks. In this experiment, we adopt the average accuracy (AA) to quantify the performance: $AA = \frac{\sum_{i=1}^{|\mathcal{T}|} \mathbf{M}_{|\mathcal{T}|,i}^{acc}}{|\mathcal{T}|}$, where $\mathbf{M}^{acc} \in \mathbb{R}^{|\mathcal{T}| \times |\mathcal{T}|}$ denotes the accuracy matrix and $\mathbf{M}_{i,j}^{acc}$ denotes model's accuracy on task $\mathcal{T}_j$ after learning task $\mathcal{T}_i$. Under the setting of AP=AA, we have $AF = \frac{\sum_{i=1}^{|\mathcal{T}|-1} \mathbf{M}_{|\mathcal{T}|,i}^{acc} - \mathbf{M}_{i,i}^{acc}}{|\mathcal{T}|-1}$. All the experiments are repeated 5 times, and the results are presented by means and standard deviations.

**Baselines and backbones.** We select our baselines from CGLB [47] including Elastic Weight Consolidation (EWC) [18], Learning without Forgetting (LwF) [20], Memory Aware Synapses (MAS) [1], Gradient Episodic Memory (GEM) [23], Experience Replay GNN (ER-GNN) [51], Topology-aware Weight Preserving (TWP) [22] and Sparsified Subgraph Memory (SSM) [49]. Additionally, we adopt joint training [6] as an approximation upper bound for

model performance, and fine-tuning (without taking any measures against forgetting) as an approximation of the lower bound [10]. To validate the generalizability of the CGL methods, we implemented each CGL method on three mainstream GNN backbones: Graph Convolutional Networks (GCNs) [17], Graph Attention Networks (GATs) [35], and Graph Isomorphism Networks (GINs) [41]. Further implementation details are presented in *Appendix*.

### 4.2 Comparisons with State-of-the-arts

We compare the performance of our method with other baselines on four public datasets. In order to introduce more randomness, we design two versions of our method based on different sampling methods: **FTF-ER-det.** in Table 2 refers to the FTF-ER method utilizing deterministic sampling, while **FTF-ER-prob.** presents utilizing probability distribution sampling. For a detailed description of these two sampling methods, please refer to Section 3.3.

The experimental results presented in Table 2 indicate that despite our proposed FTF-ER retains only subgraphs composed of a small number of nodes, it outperforms the current state-of-the-art method in the CGL field and performs similarly to the joint training method. This indicates that our FTF sampling strategy selects the most valuable nodes from the complete training dataset. Further analysis of the experimental data reveals that the vast majority of CGL methods perform poorly under the class-IL setting, with our FTF-ER significantly outperforming them. For the SSM method designed for the class-IL setting, our FTF-ER also outperforms by 1% to 3.6% in terms of AA and exhibits even better performance on AF, surpassing by 2% to 7.1%.

Furthermore, Figure 4 shows the decrease of AA for each ER-based CGL method as the number of tasks increases. Consistent with the results in Table 2, our FTF-ER method exhibits performance close to the upper bound throughout the entire learning process on all four datasets. This further indicates the effectiveness of our method in alleviating catastrophic forgetting issues. It is noteworthy that on the OGB-Arxiv dataset, compared to the SSM and the Joint method, our FTF-ER exhibits larger fluctuations in performance across task streams. This may be related to the global topological information aggregated by HPS and the homophily [50] of the dataset. For datasets with high homophily, such as OGB-Arxiv, collecting local topological information can achieve more stable performance than collecting global topological information.

### 4.3 Ablation Studies

#### 4.3.1 *Effect of Components*. To demonstrate the effectiveness of our proposed node sampling method that integrates feature-level and topological-level information, we construct three variants of our method and compare their performance with the complete FTF-ER on the OGB-Arxiv dataset. Table 3 summarizes the ablation experimental results of FTF-ER. We observe that all three sampling methods utilizing additional information outperformed random sampling, demonstrating that GraNd and HPS can accurately measure node importance. Furthermore, the best performance of the FTF (i.e. GraNd + HPS) sampling method, which combines the two scores, indicates that the fusion of feature-level information and topological-level information surpasses the approach of using only one of these types of information. However, it is noteworthy that

Table 2: Performance comparisons on 4 datasets under the class-IL setting (↑ higher indicates better performance).

| CGL Methods | Amz Comp. | | Corafull | | OGB-Arxiv | | Reddit | |
|---|---|---|---|---|---|---|---|---|
| | AA/% ↑ | AF/% ↑ | AA/% ↑ | AF/% ↑ | AA/% ↑ | AF/% ↑ | AA/% ↑ | AF/% ↑ |
| Fine-tune (soft lower bound) | 19.4±0.0 | -99.7±0.2 | 3.2±0.2 | -95.7±0.1 | 4.9±0.0 | -84.6±2.0 | 5.3±0.9 | -94.3±2.1 |
| EWC [18] | 19.4±0.0 | -99.6±0.2 | 36.2±9.1 | -60.1±9.3 | 5.2±0.1 | -92.0±0.2 | 12.3±2.8 | -91.6±2.9 |
| LwF [20] | 19.4±0.0 | -99.6±0.2 | 3.3±0.2 | -95.3±0.3 | 4.9±0.1 | -83.7±2.8 | 7.7±1.1 | -87.7±1.8 |
| MAS [1] | 31.1±11.1 | -46.4±24.8 | 28.6±5.2 | -60.2±6.9 | 6.9±1.5 | -23.5±10.9 | 13.1±4.9 | -16.8±4.1 |
| GEM [23] | 19.7±0.8 | -99.0±1.1 | 11.3±2.3 | -85.1±2.6 | 5.0±0.1 | -88.5±0.8 | 18.0±0.9 | -84.4±1.0 |
| ER-GNN [51] | 27.5±5.1 | -88.9±6.4 | 10.0±4.3 | -80.7±4.7 | 7.5±0.8 | -82.1±1.2 | 18.0±0.9 | -84.4±1.0 |
| TWP [22] | 19.3±0.0 | -99.6±0.1 | 42.2±5.0 | -54.8±5.3 | 11.5±1.0 | -75.8±4.1 | 9.7±1.6 | -91.9±2.0 |
| SSM [49] | 93.6±0.8 | -6.5±1.1 | 76.4±0.3 | -10.6±0.4 | 54.7±2.6 | -13.8±2.1 | 93.9±1.0 | -4.6±1.2 |
| Joint (soft upper bound) | 97.8±0.1 | -0.9±0.0 | 83.0±0.1 | -2.1±0.2 | 58.8±0.3 | -12.7±0.5 | 98.3±0.3 | -0.4±0.2 |
| **FTF-ER-det. (Ours)** | **94.6±0.8** | **-4.5±1.1** | 77.5±0.1 | **-3.5±0.3** | **58.3±0.6** | **-10.5±0.5** | 95.2±0.7 | -2.9±0.8 |
| **FTF-ER-prob. (Ours)** | 93.9±1.4 | -5.3±1.9 | **77.9±0.1** | -7.3±0.2 | 57.0±1.3 | -11.8±1.7 | **95.7±0.6** | **-2.3±0.5** |

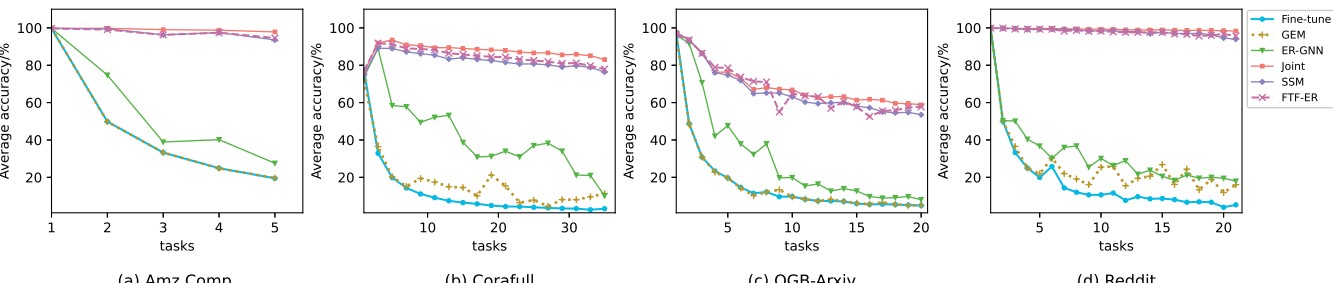

(a) Amz Comp.    (b) Corafull    (c) OGB-Arxiv    (d) Reddit

Figure 4: Evolution of the AA throughout the learning process on the task sequences of four datasets.

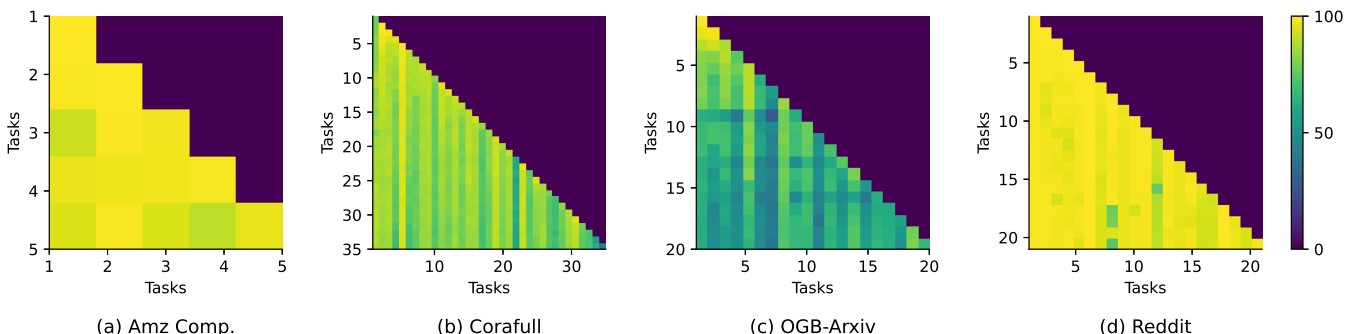

(a) Amz Comp.    (b) Corafull    (c) OGB-Arxiv    (d) Reddit

Figure 5: Visualization of the performance matrices of our FTF-ER method across four datasets.

Table 3: Ablation study on FTF-ER components.

| GraNd | HPS | Dataset | AA/% ↑ | AF/% ↑ |
|---|---|---|---|---|
| ✘ | ✘ | OGB-Arxiv | 50.9±1.9 | -15.0±1.2 |
| ✔ | ✘ | OGB-Arxiv | 52.3±0.9 | -12.5±1.0 |
| ✘ | ✔ | OGB-Arxiv | 54.6±0.8 | **-7.0±1.6** |
| ✔ | ✔ | OGB-Arxiv | **58.3±0.6** | -10.5±0.5 |

Table 4: Comparisons on feature-level node selection.

| Methods | Amazon Computers | |
|---|---|---|
| | AA/% ↑ | AF/% ↑ |
| FTF-ER (w/ Random) | 91.0±3.5 | -9.2±4.4 |
| FTF-ER (w/ MF) | 65.9±4.6 | -41.1±5.7 |
| FTF-ER (w/ CM) | 85.8±5.1 | -15.5±6.6 |
| **FTF-ER** | **94.6±0.8** | **-4.5±1.1** |

the performance of FTF-ER (HPS only) is superior to that of the FTF-ER (GraNd + HPS) on the AF, while it is inferior on the AA. This suggests that topological information is more effective in enhancing the model's memorization ability on the OGB-Arxiv dataset, and a

model that integrates both types of information can provide more stable classification effectiveness.

*4.3.2* ***Effect of GraNd.*** To demonstrate the effectiveness of gradient norm scores, we design three variants of FTF-ER, using random

**Table 5: Comparisons on topological-level node selection.**

| Methods | Amazon Computers | |
|---|---|---|
| | AA/% ↑ | AF/% ↑ |
| FTF-ER (w/ Random) | 93.5±2.0 | -5.6±2.2 |
| FTF-ER (w/ Random neighbor) | 87.7±1.7 | -13.3±2.3 |
| FTF-ER (w/ Degree neighbor) | 87.8±2.3 | -13.3±2.9 |
| **FTF-ER** | **94.6±0.8** | **-4.5±1.1** |

**Table 6: Comparisons of memory and time overheads of various ER-based CGL methods. * denotes that the methods utilize topological information.**

| Methods | Corafull | | |
|---|---|---|---|
| | AA/% ↑ | Buffer Memory/MB ↓ | Training Time/sec ↓ |
| ER-GNN[51] | 10.0 | **128.65** | 1530.16 |
| SSM[49] * | 76.4 | 195.05 | 3614.25 |
| GEM[23] | 11.3 | 625.84 | 6176.56 |
| **FTF-ER** * | **77.9** | 128.94 | **1242.74** |

scores, mean of feature (MF) scores and coverage maximization (CM) scores (both of MF and CM are proposed in ER-GNN [51]) to calculate the feature-level importance of nodes. Table 4 presents that our original method achieves the best performance in both AA and AF metrics. This further demonstrates the rationale of choosing GraNd as the method for sampling feature-level information. Interestingly, **FTF-ER (w/ Random)** achieves the second-best performance, indicating that the ER-GNN method, which performs well in the task-IL setting, does not have the capability to generalize to the challenging class-IL setting.

*4.3.3* **Effect of HPS**. To showcase the efficacy of Hodge potential scores, we design three additional variants for the method of extracting topological information to extract node topology information: random scores, random neighbor sampling, and degree-based neighbor sampling algorithms (both neighbor sampling algorithms are proposed in SSM [49]). Table 5 demonstrates that our original method achieves the best performance. This reveals that the global topological information collected by HPS is more helpful for node classification tasks compared to the local topological information collected through neighbor sampling used by SSM. Consequently, this further illustrates the rationale behind our choice of using HPS as a method for topological-level information sampling.

### 4.4 Computational Overhead

To demonstrate that our proposed method has lower computational overhead, we perform an experiment on the Corafull dataset with a budget of 60 in order to investigate the buffer storage and training time cost associated with various ER-based CGL methods. In Table 6, we use **Buffer Memory** to indicate the size of the buffer after learning all tasks, and **Training Time** to represent the total time consumption of a training session.

Table 6 shows that our proposed method has nearly the same buffer memory overhead as ER-GNN that without providing topological information, and significantly lower than the SSM and GEM methods. This demonstrates that FTF-ER can reduce the storage overhead required for utilizing topological information. Furthermore, our FTF-ER method achieves the best overall training time among all ER-based CGL methods. This is mainly attributed to

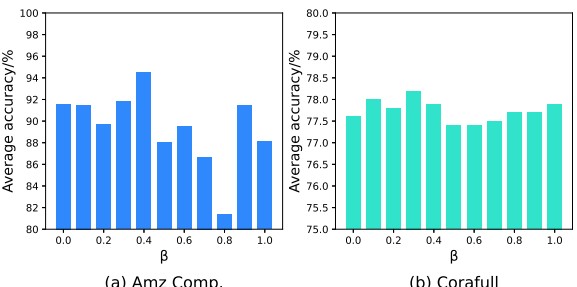

(a) Amz Comp.     (b) Corafull

**Figure 6: The influence of $\beta$ on average accuracy.**

treating topological importance calculation as a preprocessing step, thereby reducing the time overhead during runtime.

### 4.5 Sensitivity of Hyper-parameter

To analyze the sensitivity of our method to the value of the hyper-parameter $\beta$, we demonstrate the influence of $\beta$ on AA using the Amazon Computers and Corafull datasets. In this experiment, we further divide $\beta$ into [0.0, 0.1, 0.2, 0.3, 0.4, 0.5, 0.6, 0.7, 0.8, 0.9, 1.0] to observe its effects more carefully. Figure 6 displays that the value of $\beta$ has different effects on different datasets. On the Amazon Computers dataset, it leads to approximately 14% performance fluctuation, while on the Corafull dataset, it only leads to about 0.8% fluctuation. Further analysis of the experiments reveals that the peak consistently occurs at the middle position, which further confirms the effectiveness of our feature-topology fusion sampling.

### 4.6 Visualization

In order to better understand the dynamic performance of our FTF-ER method on different tasks, Figure 5 visualizes the performance matrix of average accuracy. Each cell in these matrices denotes the performance on task $j$ (column) following the learning of task $i$ (row). By looking at the matrices vertically, we can see how a specific task is gradually forgotten as training continues. By looking at the matrices horizontally, we can see how all the learned tasks perform at a given point in time. As training goes on, we notice that the colors of most tasks stay pretty much the same, matching the smooth curve of the FTF-ER method seen in Figure 4. This further validates the effectiveness of our proposed method in alleviating catastrophic forgetting issues of CGL.

## 5 CONCLUSION

In this paper, we propose FTF-ER to alleviate the catastrophic forgetting problem of CGL. From an overall perspective, FTF-ER proposes a highly complementary solution to fuse feature and topological information, thereby fully utilize the comprehensive graph data. By leveraging Hodge decomposition on graphs, we calculate the topological importance of nodes without additional storage space, and obtain more accurate global topological information compared to local neighbor sampling. We achieve state-of-the-art performance on accuracy and time efficiency in the challenging class-incremental learning setting, while maintaining comparable buffer storage costs to topology-agnostic methods. Despite the promising performance, FTF-ER may exhibit inconsistent performance on graphs with significant homophily gaps. In future studies, we will devote efforts toward adapting FTF-ER to handle heterophilic graphs.

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
