# OpenReview forum: "FTF-ER: Feature-Topology Fusion-Based Experience Replay Method for Continual Graph Learning"
_acmmm.org/ACMMM/2024/Conference — MM2024 Poster_

### Official Review · Reviewer_BBwy · 2024-05-22

**Rating:** 5
**Confidence:** 3

**Summary:**

This paper is about continual graph learning. The author proposes a novel method called Feature-TopologyFusion-based Experience Replay (FTF-ER) to effectively mitigate the catastrophic forgetting issue with enhanced efficiency. The model proposes the Hodge potential score (HPS) to capture global topological information without adding neighboring nodes to the buffer. Extensive experiments on four mainstream graph datasets verify the effectiveness of the model.

**Strengths:**

1. This paper is highly innovative and effectively alleviates the catastrophic forgetting problem in continual graph learning. The research's experiments are sufficient, and the paper is highly readable.
2. The model proposed in this paper effectively utilizes global topological information, and its performance on the data set is better than the baseline model. It has considerable time complexity and computational overhead, and the overall model is interpretable.

**Limitations:**

1. Please provide the code for the proposed model.
2. How are the hyperparameter settings determined on each dataset, especially whether there is a basis for determining the buffer size?
3. Why can utilizing local information in highly homogeneous data sets achieve more stable performance than global information? Does this prove that relying solely on global information leads to the model's limitations?
4. In Figure 6, it is best to add the influence of 𝛽 on the average accuracy of the other two datasets.

**Suitability:**

2

---

### Official Review · Reviewer_ndFG · 2024-05-25

**Rating:** 4
**Confidence:** 2

**Summary:**

This papers targets the continual graph learning and that the existing works do not fully consider the global topological information. Accordingly, this work proposes a Hodge potential score to evaluate the importance of the nodes based on the entire graph.

**Strengths:**

1. The motivation and solution is clearly introduced.

2. The target problem of continual graph learning is an important but not well explore area.

3. Experiments are conducted on several large datasets, and the results are promising.

**Limitations:**

1. Some expressions are unclear enough. E.g. what is 'more comprehensive graphs' in the abstract. This should be related to the 'global topological information'.

**Suitability:**

2

---

### Official Review · Reviewer_9NTS · 2024-05-26

**Rating:** 4
**Confidence:** 3

**Summary:**

The paper titled "FTF-ER: Feature-Topology Fusion-Based Experience Replay Method for Continual Graph Learning" introduces a novel approach to continual graph learning (CGL). The authors propose a method that combines feature-level and topological information to enhance experience replay and mitigate catastrophic forgetting. The approach utilizes gradient norm scores (GraNd) for feature importance and introduces the Hodge Potential Score (HPS) to capture global topological importance. The method demonstrates improved performance on multiple datasets, showing better accuracy and reduced memory overhead compared to state-of-the-art techniques.

**Strengths:**

Novel Combination: The integration of feature-level and global topological information for experience replay is innovative and provides a comprehensive view of node importance.
HPS Introduction: The introduction of Hodge Potential Score (HPS) to calculate global topological importance is a significant contribution, offering a new perspective on node importance in graphs.
Performance Improvement: The proposed method outperforms existing state-of-the-art methods in terms of accuracy and memory efficiency on various datasets.
Detailed Evaluation: Extensive experiments and ablation studies validate the effectiveness of the proposed method, demonstrating its robustness across different scenarios.

**Limitations:**

Complexity of Implementation: The combination of GraNd and HPS may increase the complexity of implementation, potentially requiring significant computational resources and expertise.
Limited Scope: The evaluation is primarily focused on graph node classification tasks. Extending the experiments to other types of graph-based tasks could provide a more comprehensive assessment of the method's applicability.
Potential Overhead: While the method shows reduced memory overhead, the preprocessing step for HPS calculation might introduce additional computational costs that are not thoroughly discussed.
Hyper-parameter Sensitivity: The sensitivity analysis indicates that performance can fluctuate based on the value of the hyper-parameter β, suggesting that careful tuning is necessary for optimal results.

**Suitability:**

2

---

### Meta-Review · Area_Chair_7ccH · 2024-06-30

**Recommendation:** Accept (Poster)
**Confidence:** 4

**Metareview:**

The paper introduces a novel approach to continual graph learning (CGL) by proposing a method that combines feature-level and topological information to enhance experience replay and mitigate catastrophic forgetting. Specifically, the authors propose a complementary approach to incorporate feature and global topological information and present the Hodge Potential Score (HPS) module to calculate node importance through Hodge decomposition. Despite the necessary need for hyper-parameter tuning, most reviewers agreed with the contribution of this paper to the community and the AC recommends to accept this submission as a poster.